# Proteomic Analysis Reveals the Effects of Different Dietary Protein Levels on Growth and Development of Jersey-Yak

**DOI:** 10.3390/ani14030406

**Published:** 2024-01-26

**Authors:** Guowu Yang, Rongfeng Dai, Xiaoming Ma, Chun Huang, Xiaoyong Ma, Xinyi Li, Yongfu La, Renqing Dingkao, Ji Renqing, Xian Guo, Ta Zhaxi, Chunnian Liang

**Affiliations:** 1Key Laboratory of Animal Genetics and Breeding on Tibetan Plateau, Ministry of Agriculture and Rural Affairs, Key Laboratory of Yak Breeding Engineering of Gansu Province, Lanzhou Institute of Husbandry and Pharmaceutical Sciences, Chinese Academy of Agricultural Sciences, Lanzhou 730050, China; xueshengyangguowu@163.com (G.Y.); rongfengdai123@163.com (R.D.); maxiaoming@caas.cn (X.M.); johnchun825@163.com (C.H.); abdullah33@163.com (X.M.); lixinyi9804@163.com (X.L.); layongfu@caas.cn (Y.L.); guoxian@caas.cn (X.G.); 2College of Life Sciences and Engineering, Northwest Minzu University, Lanzhou 730106, China; 3Animal Husbandry Station, Gannan Tibetan Autonomous Prefecture, Hezuo 747099, China; dingkao08@163.com; 4Zogemanma Town Animal Husbandry and Veterinary Station, Hezuo 747003, China; rqingji@163.com; 5Qilian County Animal Husbandry Veterinary Workstation, Haibei Prefecture, Qilian 810400, China

**Keywords:** dietary protein levels, Jersey-yak, proteomic, LL muscle, TMT

## Abstract

**Simple Summary:**

The aim of our study was to investigate the effects of dietary supplementation with different protein levels after grazing on growth performance as well as on the proteomics of the LL muscle of Jersey-yak. The results showed that a certain amount of crude protein supplementation to Jersey-yak at the end of grazing significantly improved growth performance and increased their economic efficiency. In addition, we analyzed the proteomic differences in the longest dorsal muscle of Jersey-yak. From our analyses, we identified differences in the abundance of 434 proteins involved in pathways related to muscle growth and development and energy metabolism, among others. Several of the proteins enriched in these pathways, such as MYH8, are associated with myoblast development and differentiation as well as muscle development. In conclusion, the results of this study provide proteomic insights into the different feeding patterns of yak crossbred progeny, suggesting that a greater economic income can be achieved by improving their feeding conditions.

**Abstract:**

Jersey-yak is a hybrid offspring of Jersey cattle and yak (*Bos grunniens*). Changing the feeding system of Jersey-yak can significantly improve its growth performance. In this study, tandem mass tag (TMT) proteomics technology was used to determine the differentially expressed proteins (DEPs) of the *longissimus lumborum* (LL) muscle of Jersey-yak fed different protein levels of diet. The results showed that compared with the traditional grazing feeding, the growth performance of Jersey-yaks was significantly improved by crude protein supplementation after grazing. A total of 3368 proteins were detected in these muscle samples, of which 3365 were quantified. A total of 434 DEPs were identified. Through analyses, it was found that some pathways related to muscle growth and development were significantly enriched, such as Rap1 signaling pathway, mTOR signaling pathway, and TGF-beta signaling pathway. A number of DEPs enriched in these pathways are related to muscle cell development, differentiation, and muscle development, including integrin subunit alpha 7 (ITGA7), myosin heavy chain 8 (MYH8), and collagen type XII alpha 1 chain (COL12A1). In conclusion, the results of this study provide insights into the proteomics of different feeding patterns of Jersey-yak, providing a stronger basis for further understanding the biological mechanism of hybrid varieties.

## 1. Introduction

Yak (*Bos grunniens*) is the main large mammal in the Qinghai–Tibet Plateau and adjacent high-altitude areas. More than 14 million yaks provide necessary meat and milk resources for Tibetans and other nomads in the plateau environment [1,2]. Because yaks live in high-altitude areas of 2000–5000 m, extremely harsh natural environmental conditions such as low temperature and low oxygen lead to a slow growth [3]. People use frozen semen of high-quality cattle such as Jersey cattle and Angus cattle to hybridize yaks through artificial insemination technology to achieve the purpose of improving the production performance of yaks [4]. Jersey-yak is a hybrid offspring of Jersey cattle and yak (a hybrid offspring of female yaks artificially inseminated with high-quality frozen semen of Jersey cattle). The production performance of the hybrid progeny has shown obvious heterosis [5]. Hybridization improves the growth performance of yak, making it adaptable to a plain environment [6]. In addition, the body length, improvement, and meat nutrition index of Jersey-yak are better than other beef cattle [7]. The cold weather conditions make the vegetation of the Qinghai–Tibet Plateau grassland in the dry season from November to May of the next year. Grasses germinate in early May, and the biomass peak appears in late August to early September [8]. During the 7-month withering period, the nutrient composition and yield of forage grass decrease sharply, meaning grazing alone cannot meet the nutritional needs of large livestock such as yak and its hybrid offspring. Yaks and their hybrid offspring have a low growth performance and even die due to insufficient nutrient intake, which eventually leads to a decrease in the economic benefits of herdsmen [9,10]. Under traditional grazing management, feed conversion efficiency is very low [11]. These results indicate that during the withering period, a reasonable crude protein level in the diet has great potential to improve the productivity of yaks and their hybrid offspring through a certain amount of supplementary feeding, and it can appropriately reduce the economic losses of herdsmen.

*Longissimus lumborum* (LL) muscle is one of the important representatives of animal muscle tissue, which is closely related to the growth and development of individual skeletal muscle, intramuscular fat content, and muscle tenderness [12]. In recent years, more researchers have explored and elucidated the molecular mechanism involved in the development of LL muscle using proteomics technology. Through the proteomic analysis of the LL muscle of yaks with different feeding methods, it was found that compared with traditional grazing, supplementary feeding significantly improved the growth performance such as average daily gain and carcass weight, and 312 differentially abundant proteins were screened out. A comprehensive analysis found that it was significantly enriched in classical signaling pathways such as glutamate metabolism and PPAR signaling pathway [13]. In order to find protein markers related to beef tenderness, 43 proteins were found through quantitative proteomics, and it was concluded that gluconeogenesis, glycolysis, and citric acid cycle were the key pathways affecting the tenderness of Piedmont beef [14]. In order to study the mechanism of yak adapting to a hypoxic environment, a quantitative phosphoproteomics analysis was performed on the muscle tissues of Gannan yak and Yushu yak at different altitudes. A total of 475 differentially expressed proteins were identified, and 26 phosphorylated proteins aggregated in energy metabolism and hypoxia adaptation were screened. Differences in protein phosphorylation levels may play a crucial role in regulating yak muscle adaptation to hypoxia [15].

Muscles are the primary site for the conversion of chemical energy into mechanical energy, which is essential for muscle contraction and the maintenance of muscle functional integrity [16]. Studies have shown that the growth traits and muscle yield of livestock are directly related to meat quality and feeding methods [17,18]. There are many factors affecting the quality of animal meat, among which nutrient (crude protein) intake is one of the very important factors. The results show that the slaughter performance and meat quality of livestock can be significantly improved through fattening [19,20]. However, there are limited studies on the muscle development and changes in the feeding system of Jersey-yak. In this study, tandem mass tag (TMT) proteomics technology was used to analyze the growth traits of Jersey-yak under different protein-level diet feeding modes, which provided necessary insights for muscle development and promoted the improvement of the productivity of Jersey-yak.

## 2. Materials and Methods

### 2.1. Ethics Approval

All the animal experiments were approved by Lanzhou Institute of Husbandry and Pharmaceutical Sciences of the Chinese Academy of Agricultural Sciences (CAAS) with the approval number: No. 1610322020018.

### 2.2. Experimental Animals and Management

This experiment was conducted in Yangnuo Yak Breeding Professional Cooperative in Xiahe County, Gannan Tibetan Autonomous Prefecture (35.01 °E, 102.57 °N, altitude 3000 m), from December 2021 to April 2022. In this experiment, 18 healthy, 6-month-old male Jersey-yaks (bred through artificial insemination of yaks with frozen semen of the same batch of Jersey cattle) were selected. A total of 18 Jersey-yaks were randomly assigned to three experimental groups based on their initial body weights using R software (v.4.1.2) [4]. The treatments of the three groups were as follows: (1) no supplementary feeding (as a control group, low protein level, LP); (2) supplementary low-protein diet (crude protein content: 15.16%; medium protein level, MP); (3) supplementary high-protein diet (crude protein content: 17.90%; high protein level, HP). The experiment consisted of two phases: a 15-day pre-test period and a 120 (4 months) day formal test period. LP, MP, and HP groups were grazed at 9:00 a.m. and ended at 18:00 p.m. In this experiment, 18 Jersey-yaks were grazed in the same natural grassland and reared in separate columns after grazing. MP group and HP group were fed with low-protein-level diet and high-protein-level diet, respectively, after grazing. The supplemental amount was calculated according to 1.2% of the body weight of Jersey-yak in the low-protein supplemental group and the high-protein supplemental group each month, and the supplemental amount of each month was adjusted according to the data obtained through weighing. The specific composition and nutritional level of the two crude-protein-level supplemental diets are shown in Appendix A.

### 2.3. Weight Data Collection and Sample Collection

At the beginning of the formal experiment (20 December 2021), 18 Jersey-yaks were weighed, and the initial body weight (IBW) of all Jersey-yaks in the three treatment groups was recorded. After the formal test began, all the Jersey cattle in the three groups were weighed on the 20th of each month until the end of the formal test. The weight data recorded during the period were the fasting weight of the Jersey-yak before grazing. Total weight gain (TWG) and average daily gain (ADG) were calculated based on the body weight data recorded during the formal experiment.

At the end of the feeding experiment, three Jersey-yaks were randomly selected from each group for slaughter, and the muscle histology of the LL muscle was studied. The LL muscle tissue samples from the 12th to 13th rib of the left half of the carcass were quickly collected during slaughter, immediately stored in liquid nitrogen, and brought back to the laboratory for subsequent experiments.

### 2.4. Sample Preparation

Grind the samples with liquid nitrogen; then, add 200 μL of SDT lysate (4% SDS, 100 mM DTT (DL-dithiothreitol, sigma, Beijing, China), 150 mM Tris-HCl pH 8.0) to each sample. Incubate at 95 °C for 3 min, sonicate for 2 min, centrifuge at 16,000× *g* for 20 min at 4 °C, take the supernatant, and use the BCA kit (Bio-Rad, Hercules, CA, USA) for protein quantification.

### 2.5. Protein Digestion

Digestion of protein (200 μg for each sample) was performed according to the FASP procedure described by Wisniewski et al. [21]. Take 200 μg of each sample for enzymatic hydrolysis; the steps are as follows: for each sample, add DTT to 100 mM, boil for 5 min, and cool to room temperature. Add 200 μL UA buffer (8M Urea, 150 mM Tris-HCl, pH 8.0) and mix well. Transfer to 10 KD ultrafiltration centrifuge tube and centrifuge at 12,000× *g* for 15 min. Repeat the operation once and discard the filtrate. Add 100 μL iodoacetamide (IAA) (50 mM IAA in UA), cool to room temperature, and centrifuge. Then, add 100 μL UA buffer and centrifuge, and repeat 2 times. Add 100 μL NH4HCO3 buffer (50 mM) and centrifuge; repeat 2 times. Then, add 60 μL Trypsin buffer (6 μg Trypsin in 40 μL NH_4_HCO_3_ buffer), shake, and let it stand at 37 °C for 16–18 h. Replace with a new collection tube, centrifuge, collect the filtrate, add 0.1% trifluoroacetic acid (TFA) solution to redissolve it, and then desalt it using a Thermo desalting spin column. The filtrate was desalted using a Thermo desalting spin column, and the peptides were quantified.

### 2.6. TMT Labeling and High PH Reverse Phase (HPRP) Fractionation

An equal amount of peptide was taken separately for each sample, and the peptides were labeled using a TMT10plex labeling kit (Thermo Fisher Scientific, Shanghai, China), and each aliquot (100 μg of peptide equivalent) was reacted separately with a tube of TMT reagent. After dissolving the sample in 100 μL 0.05 M triethyl ammonium bicarbonate (TEAB) solution (pH 8.5), the peptide was labeled using the TMT kit. After labeling, the proteins were fractionated on an Agilent 1290 HPLC using a Waters XBridge BEH130 column (C18, 3.5 μm, 2.1 × 150 mm) at a flow rate of 0.3 mL/min. The pH of two buffers (buffer A: 10 mM ammonium formate, buffer B: 10 mM ammonium formate, and 90% acetonitrile) was adjusted to 10 with ammonium hydroxide. Finally, the samples were collected and combined into 10 components. The peptides of each component were dried and redissolved with 0.1% formic acid (FA) for LC-MS analysis.

### 2.7. LC-MS Analysis (TMT10plex)

An appropriate number of peptides from each sample was separated using nano-flow Easy nLC 1200 chromatographic system (Thermo Scientific, Waltham, MA, USA). The chromatographic column was balanced with 100% buffer A solution (0.1% formic acid aqueous solution). The sample was injected into the Trap Column (100 μm × 20 mm, 5 μm, C18, Dr. Maisch GmbH) and then separated using a chromatographic column (75 μm × 150 mm, 3 μm, C18, Dr. Maisch GmbH) at a flow rate of 300 nl/min. The liquid-phase separation used buffer solution B (0.1% formic acid, 95% acetonitrile, and water-mixed solution); the separation gradient was as follows: 0–2 min, B solution linear gradient from 2% to 8%; 2–71 min, 8% to 28%; 71–79 min, 28% to 40%; 79–81 min, 40% to 100%; 81–90 min, B solution concentration maintained at 100%. After peptide separation, DDA (data-dependent acquisition) mass spectrometry was performed using a Q-Exactive HF-X mass spectrometer (Thermo Scientific). The analysis time was 90 min; detection mode: positive ion; parent ion scanning range: 400–1800 m/z; primary mass spectrometry resolution: 60,000 @m/z 200; AGC target: 3 × 10^6^; primary Maximum IT: 50 ms. The secondary mass spectrometry analysis of peptides was collected according to the following methods: after each full scan, the secondary mass spectrometry (MS2 scan) of 20 highest-intensity parent ions was triggered and collected. Secondary mass spectrometry resolution: 45,000 @m/z 200; AGC target: 1 × 10^5^; secondary Maximum IT: 50 ms; MS2 Activation Type: HCD; isolation window: 1.2 m/z, with 32 as the normalized collision energy.

### 2.8. Database Searching

The final LC-MS/MS original RAW file was imported into the search engine Sequest HT in Proteome Discoverer software (v.2.4, Thermo Scientific) for database retrieval. The database used was Bosgru_v3.0 [15]. The cutting enzyme used was Trypsin/P, allowing for the omission of up to two cleavage sites. The initial search setting was determined to be 10 ppm, and the mass tolerance for fragment ions was limited to 20 ppm (MS/MS Tolerance: 0.02 Da). TMT10plex (N-term), Carbamidomethyl (C), and TMT10plex (K) were used as fixed modifications, and Acetyl (Protein N-term) and Oxidation (M) were used as variable modifications. False discovery rate (FDR) was set at 1%. Isobaric labels were used to quantify TMT 10plex.

### 2.9. Bioinformatics Analysis

Perseus software and R statistics were used for data analysis. The fold-change (FC) of >1.20 or <0.83 and *p* values < 0.05 were used to screen differentially expressed proteins (DEPs). Kyoto Encyclopedia of Genes and Genomes (KEGG) pathway (https://www.genome.jp/kegg/, v.109.0, accessed on 1 July 2023) and Gene Ontology (GO) annotations (http://www.geneontology.org/, v.42.442, accessed on 1 July 2023) were used to analyze DEPs. GO and KEGG enrichment analyses were performed using Fisher’s exact test, and FDR correction of multiple tests was performed. Clustering heatmaps were generated using OECloud tools, a free online data analysis platform (https://cloud.oebiotech.com, v.1.26, accessed on 5 July 2023). Gene set enrichment analysis (GSEA v.4.1.0) was used to analyze the signaling pathways enrichment in different groups [22]. K-means clustering was performed on all DEPs using an online data analysis software (http://www.bioinformatics.com.cn/, accessed on 6 July 2023). Construction of protein–protein interaction (PPI) networks was also conducted by using the STRING v11.5 (https://cn.string-db.org/, accessed on 6 July 2023) database with the Cytoscape (v.3.9.1) software, which excludes networks showing ≤2 nodes [23,24].

### 2.10. Western Blotting

Muscle tissue blocks were washed 2–3 times with pre-cooled PBS, cut into small pieces, and placed in a homogenizer tube, adding 10 times the tissue volume of RIPA lysate (G2002, Servicebio, Wuhan, China) and PMSF (Phenylmethanesulfonyl fluoride, Servicebio, China) protease inhibitor (G2008, Servicebio, China), which was homogenized. The homogenate tube was taken out and placed on ice for lysis for 30 min. The tissue was shaken every 5 min to ensure complete lysis: 12,000× *g* rpm, 4 °C, centrifuged for 10 min, collect the supernatant, which is the total protein solution. After SDS-PAGE electrophoresis, the protein extract was imprinted on the PVDF (0.45 μm, WGPVDF45, Servicebio, China) membrane. The primary antibodies used in Western blotting were as follows: mouse monoclonal anti-GAPDH (ServicebioGB15002, China, 1:2000), rabbit polyclonal anti-ALB (ServicebioGB11319, China, 1:1000), rabbit polyclonal anti-PHKA1 (ServicebioGB113060, China, 1:1000), rabbit polyclonal anti-TPM2 (Bioss1243R, China, 1:1000), and rabbit polyclonal anti-ACTN1 (CohesionCPA3466, China, 1:1000). Secondary antibodies were incubated using HRP conjugated Goat Anti-Rabbit IgG (H + L) (ServicebioGB23303, China, 1:3000). Protein bands were visualized via chemiluminescence using Hypersensitive ECL Chemiluminescence Kit (Servicebio G2020-50ML, China). All images were analyzed using AIWBwell^TM^ analysis software (Servicebio, China).

### 2.11. Statistical Analysis

One-way ANOVA and LSD multiple comparisons were performed on all data using SPSS 25.0 software (IBM, Armonk, NY, USA). The results were expressed as mean ± SE.

## 3. Results

### 3.1. Effects of Different Protein Levels on Body Weight of Jersey-Yak

The effect of crude protein level in the supplementary diet on the body weight of Jersey-yak is shown in Table 1. There was no significant difference in body weight between the three protein levels in the initial test and the first month (*p* > 0.05). After the start of the formal experiment, the weight difference between the three groups in the second month was significant (*p* < 0.05). The final body weight after the test was significantly different among the three groups (*p* < 0.01). After the end of grazing, the weight of the MP group and HP group was significantly higher than that of the LP group from the second month (*p* < 0.05). The average daily gain (ADG) and total weight gain (TWG) in the MP group and HP group were significantly higher than those in the LP group (*p* < 0.05); ADG and TWG were not significantly different between the MP group and HP group (*p* > 0.05).

During the formal experiment, the effects of supplementary diets with different protein levels on the body weight of Jersey-yak were observed as shown in Figure 1. In the comparison of body weight, the body weight of the MP group and HP group was significantly higher than that of the LP group. It can be seen from the figure that the body weight of Jersey-yak in the MP group and HP group had an increasing trend, but the increasing trend gradually decreased. Similarly, the increasing trend of the LP group without supplementary feeding also gradually decreased, but the weight of the LP group showed a negative growth in the last month.

### 3.2. Quantitative Proteomics-Based Analysis of LL Muscle Protein in Jersey-Yak

The LL muscle of Jersey-yak was analyzed using LC-MS/MS technology, and 100,380 available effective spectra were obtained. A total of 22,443 specific peptides were detected from these spectral analyses, and 3368 proteins were detected, of which 3365 proteins were quantified (Appendix A). The predicted molecular weight (MW) values of these proteins varied greatly, ranging from 3.9 to 797.2 kDa (Appendix A). The detected peptide length (Appendix A) and the number distribution of specific peptides (Appendix A) were evaluated. According to the expression of proteins in different samples, principal component analysis (PCA) was performed on the quantitative proteins in LP, MP, and HP muscle samples. The score and correlation proved that the samples had good clustering and reliable data (Appendix A). The protein molecular weight and isoelectric point distribution (Appendix A) and protein coverage distribution (Appendix A) were also evaluated and analyzed.

### 3.3. Differentially Expressed Protein Analyses

We compare the differences in protein expression between three muscle samples. For HP vs. LP, HP vs. MP, and MP vs. LP muscle samples, 150 (46 upregulated, 104 downregulated), 73 (12 upregulated, 61 downregulated), and 211 (99 upregulated, 112 downregulated) DEPs were identified, respectively (Figure 2A). For more details about these proteins, see Appendix A. The Venn diagram was used to highlight the DEPs associated with each pair of samples (Figure 2B). Overall, the whole proteome profiles of HP and MP muscle samples were the most similar, with only 73 differential proteins in the two groups. There were 36 protein overlaps between HP vs. LP and MP vs. LP muscle samples, and 26 protein overlaps between HP vs. LP and HP vs. MP muscle samples. A total of 434 DEPs were identified in these analyses, of which 1 was shared in all analyzed sample comparisons.

### 3.4. DEPs Cluster Analysis

The hierarchical clustering method was used to analyze the abundance changes in the DEPs in the LL muscle of Jersey-yak in different groups (Figure 2C). Each color in the graph corresponds to a different DEP abundance value, and the proteins showed different colors in different samples, suggesting that the same DEPs had different abundance values in the LL muscle of Jersey-yak reared at different protein levels. As can be seen in the figure, those with consistent expression abundance values are basically clustered together. The three biological replicates of each group showed similar colors, indicating that the DEPs were basically consistent in the same group of LL muscle samples.

### 3.5. Functional Analyses of Identified DEPs

These proteins were classified via different bioinformatic analyses to identify the potential biological functions of the DEPs identified when comparing these groups of Jersey-yak LL muscle fed diets with different protein levels (Figure 3A, Appendix A). GO analysis of DEPs identified through a comparison of HP vs. LP LL muscles reveals that these proteins are enriched for biological processes (BPs) including actin filament-based process and muscle system process; cellular components (CCs) including cytoplasm, actomyosin, and actin filament bundle; and molecular functions (MFs) including binding, protein dimerization activity, and catalytic activity. The GO analysis of the DEPs identified by the MP vs. LP group revealed these proteins to be enriched for BPs including glycogen metabolic process and muscle contraction; CCs including cytoplasm, mitochondrion, and cytoskeleton; MFs including binding, structural molecule activity, and protein dimerization activity. The GO analysis of the DEPs identified by the HP vs. MP group revealed these proteins to be enriched for BPs including cytoskeleton organization and muscle contraction; CCs including actin cytoskeleton, cytoplasm, and cytoskeleton; MFs including binding, catalytic activity, and protein serine/threonine kinase activity.

A KEGG pathway enrichment analysis was performed on the DEPs of the three comparison groups (Figure 3B, Appendix A). The DEPs identified in the HP vs. LP comparison were significantly enriched in proteasome, HIF-1 signaling pathway, and carbon metabolism. The DEPs identified in the HP vs. MP comparison were significantly enriched in motor proteins, lysine degradation, and relaxing signaling pathway. The DEPs identified in the MP vs. LP comparison were significantly enriched in the calcium signaling pathway, TGF-beta signaling pathway, and glucagon signaling pathway.

### 3.6. GSEA Protein Set Enrichment Analysis

The Gene Set Enrichment Analysis (GSEA) is an analysis method for proteome-wide expression profiling data, which compares proteins with predefined protein sets [25]. Unlike GO and KEGG enrichment, the results of which are related to the number of differential proteins and the setting of screening parameters of differential proteins, which have certain limitations, GSEA is not limited to differential proteins. GSEA can explain a certain pathway screened by KEGG enrichment, as well as the upregulated and downregulated proteins in this pathway; it can also predict whether this pathway is inhibited or activated. Subsequently, with HP as the control group and LP as the test group, two and eight gene sets were identified using the GSEA method, which were significantly downregulated and upregulated, respectively, in the muscle samples of the LP group (Appendix A). Notably, one of these downregulated gene sets was associated with the proteasome pathway (Figure 4A). One and eight gene sets were identified in the HP vs. MP comparison group, which were significantly downregulated and upregulated, respectively, in the muscle samples of the MP group (Appendix A). Among them, one of these downregulated gene sets was associated with a cAMP signaling pathway (Figure 4B). In total, 19 and 4 gene sets were identified in the MP vs. LP comparison group, which were significantly upregulated and downregulated, respectively, in muscle samples from the LP group (Appendix A). The downregulation of gene sets associated with the PI3K-Akt signaling pathway (Figure 4C) and VEGF signaling pathway (Figure 4D) was revealed, and the protein group related to the Calcium signaling pathway (Figure 4E) was upregulated.

### 3.7. K-Means Cluster Analysis of DEPs

In order to determine the expression patterns of differential proteins in different muscle samples, we performed k-means clustering on all DEPs. We performed a Z-score transformation on the expression values of the significant proteins (*p* < 0.05) in the ANOVA test results, and then used the Hartigan–Wong method for k-means clustering analysis. Using the K-means clustering method, according to the similarity of protein expression levels, 466 DEPs were clustered according to their expression trends and divided into four clusters (Figure 5, Appendix A). GO enrichment analysis was performed on the four clusters (Appendix A), and many BPs were found to be related to muscle growth and development. In cluster 1, proteins were enriched in muscle tissue development (SRPK3, ACTA1, CAV3, and MYH14), cytoskeleton organization (CAV3, LIMCH1, and COBL), and protein folding (PPIL1, HSPB6, AHSA1, CCT5, and CLGN). It was also found that proteins in cluster 2 were enriched in BPs of skeletal muscle contraction (HOMER1, TNNC2, MYH8, and SCN4A), glycogen metabolic process (AGL, PHKB, PHKA1, SLC37A4, and PHKA2), and regulation of muscle contraction (TNNC2, AKAP9, ATP2A1, and SCN4A). In cluster 3, proteins were enriched in the developmental process (GDI1, COL12A1, HMGB2, and TNC), response to hypoxia (MMP2, HP1BP3, and HMOX1) and muscle structure development (MECP2, LGALS1, DAG1, and TCAP), and their expression levels peaked in the LP group. BPs related to cell junction organization (SMAD3, SLK, ACTN1, and CLASP1) and actin filament organization (TPM4, SORBS3 and CLASP1) were found in cluster 4.

### 3.8. PPI Analyses

Next, a PPI analysis was performed on the DEPs identified via the MP and LP comparison groups (Figure 6). The network mainly includes proteins involved in muscle contraction, carbohydrate metabolic process, protein metabolic process, cytoskeleton organization, and protein binding. Key hub proteins critical to the stability of this network include ALB, COL2A1, COL6A2, PHKA1, MYH8, and MYBPC1.

### 3.9. Western Blotting

In order to verify the results of the above quantitative proteomics analysis, some identified DEPs were selected for a Western blotting-based confirmation of the observed expression trends detected after TMT labeling. The expression of the four selected representative proteins (ALB, PHKA1, TPM2, ACTN1) was basically consistent between TMT and Western blot analysis (Figure 7).

## 4. Discussion

Interspecific hybridization has long been used by humans to produce hybrid offspring, which are sometimes stronger or perform better than their parental lineages (heterosis or heterozygote heterosis) [26]. The production capacity of purebred cattle and hybrid cattle was studied. It was found that the slaughter age of hybrid cattle was younger, and the live weight, carcass weight, and slaughter rate were higher than those of purebred cattle [27]. The results showed that under the same feeding conditions, Simmental crossbred cattle had better meat performance and provided low-fat meat with beneficial fatty acid composition, but the meat quality was poor, while the meat color and protein content of crossbred cattle were better and higher [28]. A study of different feeding patterns on yaks showed that house-feeding can significantly improve the production performance of yaks [13]. The results showed that the growth performance of yaks could be improved by supplementing different crude protein levels after the grazing of early-weaned yaks, and the energy deficiency of female yaks in the winter feed scarcity period could be minimized [29]. Similarly, strategic feed supplementation for mature (5~13 years) yaks during forage scarcity can increase milk production and reduce calving intervals [30]. These results indicate that hybrid offspring have better production performance, and supplementary feeding can obtain higher economic benefits.

This experiment was conducted to study the different feeding modes of Jersey-yak, and to explore the effect of dietary protein level on the LL muscle of Jersey-yak. The results showed that compared with the pure grazing group, the total weight gain of Jersey-yak in the MP group and HP group increased by 15.13 and 18.23 kg, respectively. Subsequent analyses revealed that the reduced remodeling activity with muscle growth in the HP group may have been due to an excess of protein supplementation in the diet of the HP group, leading to a possible shift from muscle growth to the growth of other tissues. In contrast, the weight loss in the LP group may be related to muscle catabolism, as the LP group did not receive dietary supplementation after grazing, which may have led to weight loss.

In order to better understand the effect of dietary protein levels on the growth traits of Jersey-yak, 3368 proteins were identified in the LL muscle of Jersey-yak using the TMT proteomics technology and high-resolution LC-MS/MS analysis technology, of which 3365 were quantified effective proteins. The proteins in this study were quantified via BCA analysis, but they may be interfered with by other chemicals, such as reducing agents, chelating agents, surfactants, and so on. Therefore, we performed experimental control and quality control during the experiment to ensure the accuracy and reliability of the results. Among the differential proteins, the highest expression abundance was MYH isoform. The MYH proteins expressed in this study were mainly MYH1, 2, 3, 4, 6, 7, 8, 9, 10, 11, 13, 14, and 7B. The *MYH* gene has a common molecular function (i.e., force generation), but it has different expression patterns in terms of spatial and temporal patterns and cell types [31]. From the perspective of skeletal muscle cell type, *MYH7* is mainly expressed in slow muscle fibers, while *MYH1,2,4* are mainly expressed in fast muscle fibers; *MYH6* is mainly expressed in cardiomyocytes, and *MYH13* is mainly expressed in extraocular muscles (EOMs) [32]. Among them, *MYH7B* has also been detected in non-muscle tissues of mammals, and *MYH7B* mutations are associated with hereditary hearing loss in compound heterozygous patients [33,34]. The phylogenetic analysis of *MYH7B* showed that it could be classified as a member of myosin-II family [35]. The increase in muscle mass can be achieved mainly by increasing the number and size of fibers and (or) by converting slow muscle fibers into fast muscle fibers [36]. This is consistent with the expression of MYH1 and MYH4 proteins in the results of this experiment. The results of this experiment showed that (Appendix A) the expression abundance of MYH1 and MYH4 proteins in the HP group was significantly higher than that in the LP group (*p* < 0.05). In the EDPs of HP vs. MP comparison group, the expression of MYH2 in the MP group was downregulated, which was consistent with the above conclusion.

GO enrichment analysis showed that multiple DEPs were involved in the formation of muscle cells and muscle tissues such as muscle system process, cytoskeleton organization, muscle structure development, muscle cell differentiation, and animal organ development. Similarly, there are different metabolic processes, such as cellular catabolic process, carbohydrate metabolic process, and catabolic process. It is also involved in muscle contraction and other biological processes related to meat quality after slaughter [37]. Among the identified proteins, several proteins are related to the development, differentiation, and muscle development of muscle cells [38], namely, integrin (ITGA1, ITGA5, ITGA6, ITGA7, and ITGA2B), collagen (COL12A1, COL14A1, COL6A1, COL6A2, COL1A2, and COL3A1), coronin (CORO1A, CORO1B, CORO1C, and CORO6), myosin light chain (MYL1, MYL2, MYL3, MYL4, MYL6B, MYL6, MYL9, MYLK, and MYLPF), myosin heavy chain (MYH1, MYH2, MYH4, MYH6, MYH7, MYH8, MYH9, and MYH7B), troponin (TNNI1, TNNI3, TNNC1, and TNNC2), and some regulatory factors, indicating that the above proteins may play a certain role in the formation of muscle cells and muscle tissue. For example, in the HP vs. LP group, the expression of ITGA7, COL12A1, and CORO1B was downregulated in the HP group. In the MP vs. LP group, the expression of COL12A1, COL6A2, and COL2A1 was downregulated in the MP group, and the expression of CORO6, MYL1, and MYH8 was upregulated in the MP group. Among them, the integrin subunit alpha 7 (ITGA7) protein is mainly expressed in skeletal muscle and myocardium, and it may be involved in differentiation and migration during myogenesis. The defect of this gene is associated with congenital myopathy [39,40,41]. In addition, studies have found that ITGA7 plays an important role in the early stages of osteogenesis by upregulating the PI3K-AKT signaling pathway [39].

In our study, KEGG enrichment analysis showed that many signaling pathways were significantly associated with muscle growth, development, and energy metabolism, including TGF-beta signaling pathway, mTOR signaling pathway, Rap1 signaling pathway, carbon metabolism, calcium signaling pathway, insulin signaling pathway, and dilated cardiomyopathy (DCM) [42,43]. The transforming growth factor beta (TGF-beta) signaling pathway is a series of signal transduction processes mediated by a transforming growth factor. TGF-beta family cytokines play a variety of roles in regulating the growth, differentiation, immune response, and development of multiple-organ systems [44,45]. Studies have shown that TGF-beta receptors bind to ligands to activate Smad proteins and activate the transcriptional regulation of target gene expression through phosphorylation, as well as the control mechanism of Smad protein activity and degradation [45]. Among them, Smad proteins are a class of molecules that are directly involved in TGF-beta signal transduction and play an important role in TGF-beta signal transduction [46]. SMAD3 belongs to the receptor-regulated Smad protein, which is upregulated in the MP group. Members of the TGF-beta family initiate their cellular actions by binding to receptors with intrinsic serine/threonine kinase activity [47].

Rap1 is a Ras-like small GTPase that is mainly involved in biological functions mediated by integrin proteins, such as the control of cell adhesion, and it can be activated by many extracellular stimuli [48]. Studies have found that Rap1 plays a very important role in the formation of cadherin-based cell–cell junctions, and the Rap1 signaling pathway is also indispensable in the regulation and maintenance of cell-to-cell contacts [48]. Therefore, one of the functions of Rap1 activation may be to recruit guanine nucleotide exchange factors (GEFs) of Rac and Cdc42 proteins to the location of the cell–cell’s initial contact, thereby connecting to the actin cytoskeleton [49]. By studying the expression and localization of Rap1 protein in different stages of skeletal muscle development, the correlation of Rap1 protein development mechanism in vivo was verified. It was found that Rap1 protein was accumulated in specific muscle cell domains, which were modified by neuromuscular and tendon junctions in the early and late stages of myogenesis, indicating that Rap1 protein was regulated during myogenic differentiation [50]. In addition, the Rap 1 signaling pathway interacts with the β-adrenergic signaling pathway, a response that plays a critical function in the growth and development of skeletal muscle [51].

The mammalian target of rapamycin (mTOR) is a serine-threonine protein kinase expressed in all cells [52]. mTOR plays a role in many biological functions, such as regulating cell transcription, translation, cell growth, cell differentiation, and apoptosis [53]. The mTOR signaling pathway is critical for growth and development, mainly related to insulin signaling, growth factors, nutrition, and energy metabolism [54,55]. It is worth noting that the insulin signaling pathway is also involved in growth and metabolism and plays an important role in muscle growth and development [42]. Similarly, the study also found that the disorder of the mTOR signaling pathway may be related to the progress of cancer, diabetes, and aging process [55]. In addition, mTOR activation can promote protein translation, promote cell growth, and affect intracellular metabolism, and mTOR plays a particularly important role in metabolic organs such as liver, muscle, and adipose tissues to regulate systemic energy homeostasis [56]. In summary, among the upregulated or downregulated DEPs in the MP and HP groups, DEPs related to TGF-beta, Rap1, mTOR, and insulin signaling pathways may be regulatory proteins for the muscle growth and development of Jersey-yak.

In the HP vs. LP comparison group, significant enrichment of hypoxia-related signaling pathways in Jersey-yak was found, including DEPs of hypoxia-inducible factor 1 (HIF-1) signaling pathways such as PDHA2, HMOX1, INSR, and PRKCA. The HIF-1 signaling pathway is a transcription factor that plays a major regulatory role in the body ‘s response to hypoxic concentration or hypoxia and in oxygen homeostasis [57,58,59,60]. Under hypoxic conditions, HIF-1 acts as a major regulator of many hypoxia-inducible genes. The target gene of HIF-1 encodes a protein that increases oxygen delivery and mediates an adaptive response to oxygen deprivation. This is related to the adaptation of Jersey-yak to a hypoxic environment in the plateau area.

Next, GSEA analysis was used to further study the signal transduction pathway related to the muscle growth and development of Jersey-yak. It was found that the genome of Calcium signaling pathway was significantly upregulated in MP muscle samples. The role of Ca^2+^ signaling has been considered in each of these developmental steps, and clear evidence that there are different roles of Ca^2+^ signaling in different aspects of a wide range of species have been confirmed. The identified molecular mechanisms of Ca^2+^ involvement in muscle development is responsible for the formation of Ca^2+^ dynamics or the transduction of Ca^2+^ signals to cellular responses. In developing muscle cells, Ca^2+^ storage is essential for eliciting precise spatial and temporal patterns of Ca^2+^ signaling [61]. Studies have shown that the calcium signaling pathway plays a key role in the maintenance and regeneration of skeletal muscle at different stages of prenatal muscle development and after birth [61]. The EDPs enriched in the Calcium signaling pathway in the muscle samples of the MP group included PHKA1, PHKA2, PHKB, PHKG1, ATP2A1, ATP2A2, and PLCD4. It is worth noting that these proteins are upregulated in the MP group. Phosphorylase kinase (PHK) is a polymer of 16 subunits, 4 subunits each for α, β, γ, and δ [62], and it is mainly expressed in muscles and the liver. PHK is a regulatory protein kinase that stimulates glycogenolysis by phosphorylating and activating glycogen phosphorylase [63]. In muscles, they lead to energy production to sustain muscle contraction, and in the liver, these responses are used to maintain blood glucose [64]. Since the main storage sites of mammalian glycogen are liver and muscle, the expression of PHK subunits is considered to have tissue distribution limitations, among which phosphorylase kinase regulatory subunit alpha 1 (PHKA1) and phosphorylase kinase catalytic subunit gamma 1 (PHKG1) proteins are considered to be specifically expressed in the muscle [64], and PHKA1 is a key regulatory enzyme of glycogen metabolism [63].

The PI3K-AKT pathway is a classical cellular signaling pathway in cells that is involved in many cellular functions, including cell growth, cell differentiation, and cell motility [65,66]. The study found that the PI3K-AKT signaling pathway was essential in all stages of bone maturation, differentiation, and bone growth. The inhibition of the PI3K-AKT signaling pathway not only impairs chondrocyte differentiation but also inhibits longitudinal bone growth [67]. It shows that the PI3K-AKT signaling pathway plays a key role in the maturation and growth of Jersey-yak bone. In addition, the PI3K-AKT signaling pathway plays a critical role in the regulation of myogenesis and muscle hypertrophy in the skeletal muscle [68]. Therefore, the PI3K-AKT signaling pathway also plays an important role in the muscle growth and development of Jersey-yak. The screened differential proteins involve the COL (Collagen) family members COL2A1 and COL6A1 proteins in the PI3K-AKT signaling pathway proteins, which are involved in the formation of cell network scaffolds and play a role by encoding the α1 chain of types II and VI collagen. Among them, the COL2A1 protein may be related to the cell differentiation and organ development of Jersey-yak [69].

Albumin (ALB) protein is the most abundant protein in the blood, which is mainly involved in the regulation of plasma colloid osmotic pressure, and serves as a carrier protein for a variety of endogenous molecules, including hormones, fatty acids, metabolites, and exogenous drugs [70,71]. ALB protein plays a key role in the antioxidant capacity of blood plasma in animals and humans, rendering potential toxins harmless and removing reactive oxygen species, which are important antioxidants in the body [72,73]. The study found that ALB protein was involved in the transport of nutrients in heifers with higher intramuscular fat content, and it also revealed the enrichment of ALB protein in oxidatively active pathways [74]. A PPI analysis found that ALB and COL2A1, COL6A2, RABEP1, MME, CCDC6, and other proteins work together in the protein-binding pathway.

## 5. Conclusions

The results showed that a certain amount of supplementary feeding after the end of grazing could significantly improve the production performance of Jersey-yak. The proteins involved in the TGF-beta signaling pathway, mTOR signaling pathway, Rap1 signaling pathway, carbon metabolism, calcium signaling pathway, insulin signaling pathway, and dilated cardiomyopathy pathway may be related to the muscle growth and development of Jersey-yak. In general, the proteins related to the effect of crude protein level in the supplementary diet on the LL muscle of Jersey-yak were screened out, which provided a theoretical reference for the genetic improvement and fattening of hybrid cattle in the future.

## Figures and Tables

**Figure 1 animals-14-00406-f001:**
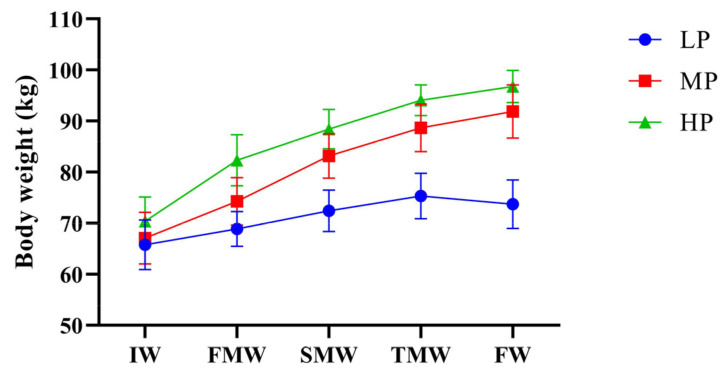
Effects of dietary protein level on body weight of Jersey-yak. IW, initial weight; FMW, first-month weight; SMW, second-month weight; TMW, third-month weight; FW, final weight. LP, low protein level; MP, medium protein level; HP, high protein level.

**Figure 2 animals-14-00406-f002:**
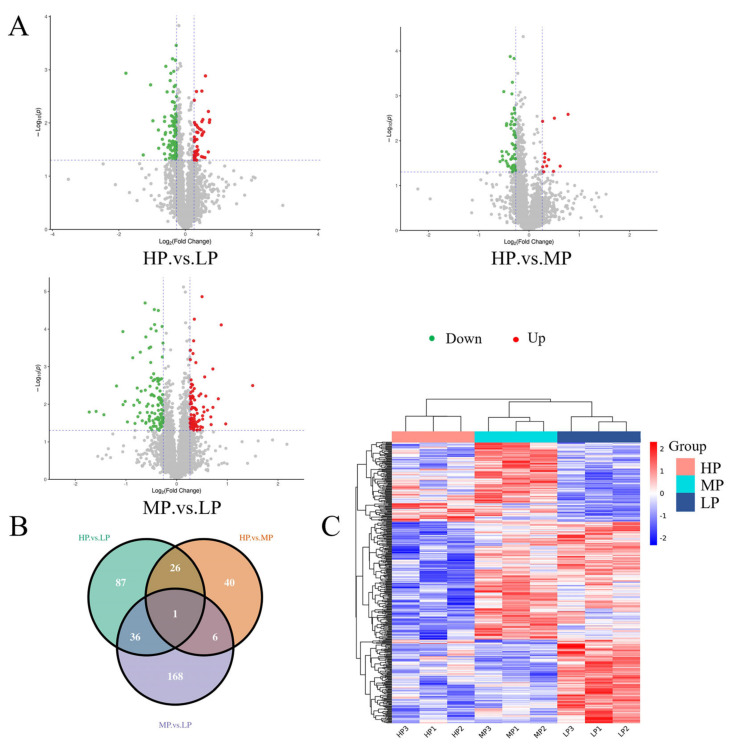
Differential protein expression patterns among samples. (**A**) Volcano plots represent differentially expressed proteins between each comparison group. Red dots and green dots correspond to upregulated and downregulated proteins, respectively. The vertical dashed lines indicate |log2FC| = 0.83 and the horizontal dashed lines indicate *p* = 0.05. (**B**) Venn diagram showing the cascading relationship of DEPs in different comparison groups (HP vs. LP, HP vs. MP, and MP vs. LP). (**C**) Cluster analysis of DEPs in LL muscle of Jersey-yak. Columns and rows represent samples and proteins, respectively.

**Figure 3 animals-14-00406-f003:**
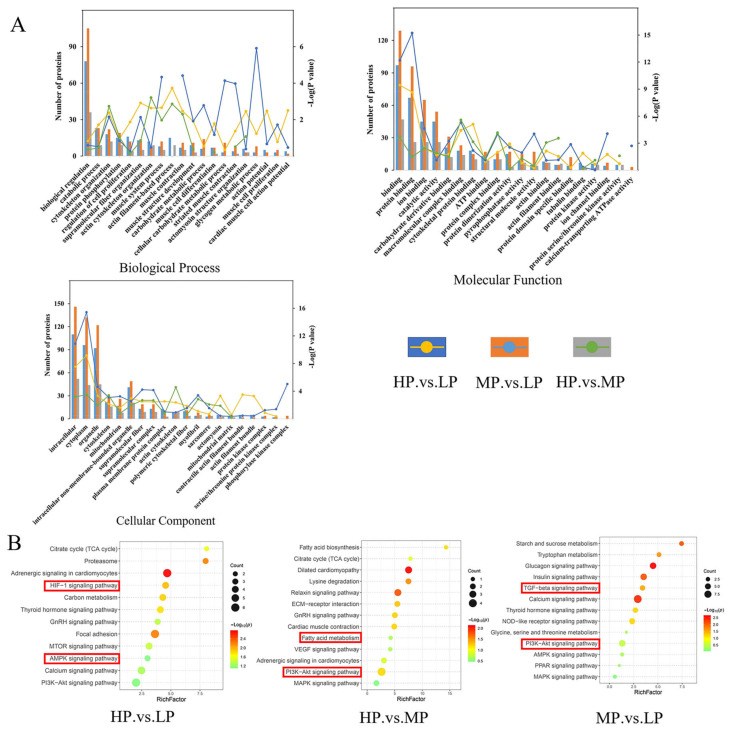
Functional enrichment analyses. (**A**) GO term enrichment for BP, CC, and MF terms for DEPs from the indicated LL muscle group comparisons. (**B**) KEGG pathway analysis of DEPs. Note: The red boxes in the diagram show the pathways that require special attention.

**Figure 4 animals-14-00406-f004:**
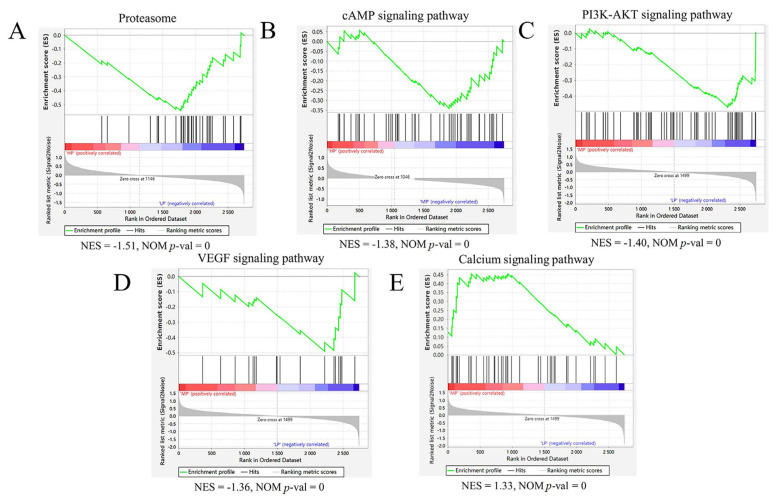
GSEA enrichment map of each comparison group’s proteome based on KEGG pathway. (**A**) The proteasome hallmark gene set was downregulated in the HP vs. LP muscle comparison. (**B**) The cAMP signaling pathway hallmark gene set was downregulated in the HP vs. MP muscle comparison. (**C**,**D**) The PI3K-AKT signaling pathway and VEGF signaling pathway hallmark gene sets were downregulated in the MP vs. LP muscle comparison. (**E**) The Calcium signaling pathway hallmark gene set was upregulated in the MP vs. LP muscle comparison. The green broken line indicates the enrichment score (ES) of the protein, and the proteins are sorted from large to small according to FC from left to right. When the ES < 0, the function is enriched at the back end of the protein ranking, indicating low protein expression, and the function is inhibited, but activated vice versa. NES = normalized enrichment score; NOM *p*-val = nominal *p* value.

**Figure 5 animals-14-00406-f005:**
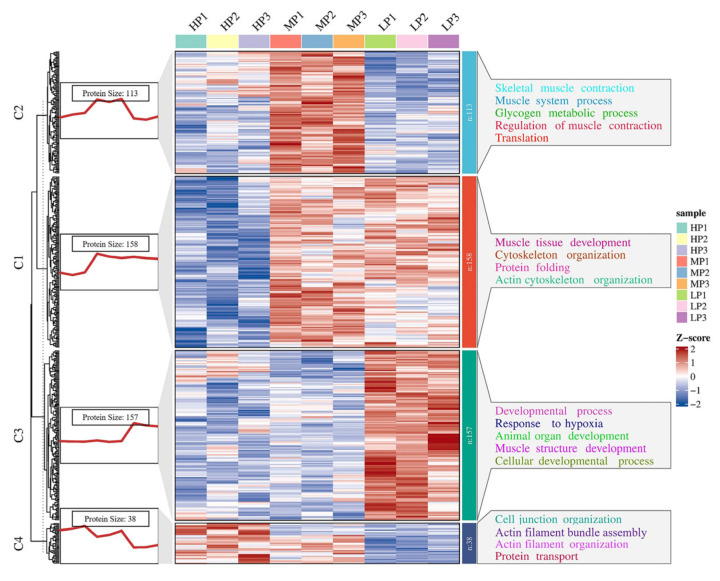
K-means cluster analysis map of significantly different proteins, k = 4. C1~4 = clusters 1~4. The red trend line on the left represents the average expression trend of all proteins in the cluster. The heatmap represents the expression pattern of all proteins in the cluster. On the right are some GO teams of differential proteins in each cluster.

**Figure 6 animals-14-00406-f006:**
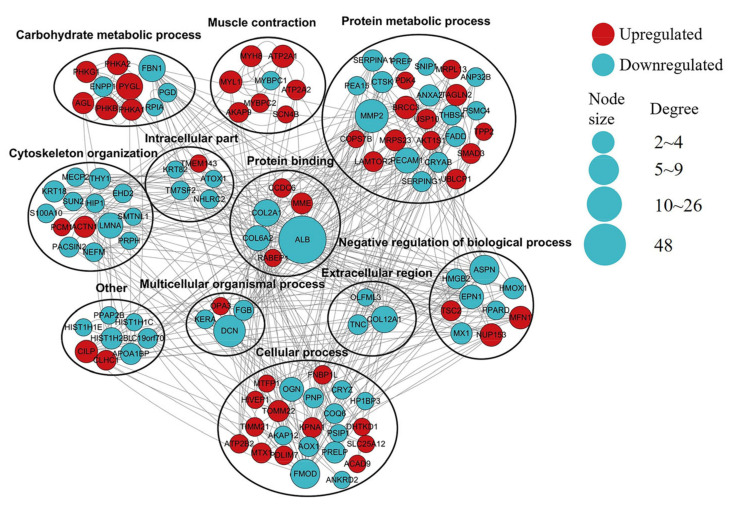
Predicted PPI network for proteins differentially expressed between MP and LP muscle samples. In the constructed network, DEPs represent nodes. The interaction between proteins was labeled as >2 connected proteins. Red indicates significant upregulation, and green indicates significant downregulation. Proteins were grouped according to known GO biological functions.

**Figure 7 animals-14-00406-f007:**
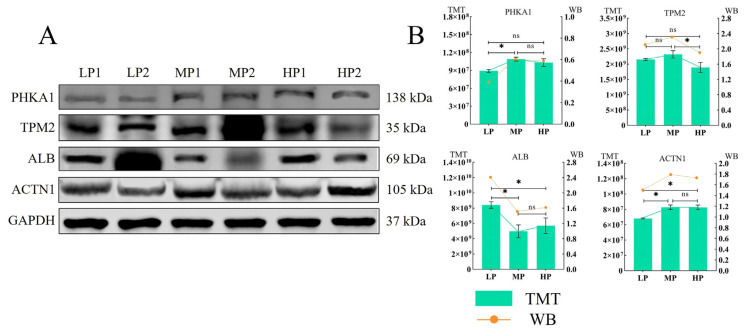
(**A**,**B**) The quantitative proteomics results of the four proteins were verified based on Western blot, and GAPDH was used as the sample control. “*” indicates *p* < 0.05, “ns” indicates that the difference is not significant.

**Table 1 animals-14-00406-t001:** Effects of dietary levels on body weight of Jersey-yak.

Item ^2^	Treatment Group (Unit: kg, Mean ± SEM ^1^)			
	LP	MP	HP	*p*-Value
Pre-test weight	64.92 ± 4.73	63.84 ± 4.54	64.84 ± 5.91	0.986
Initial weight	65.80 ± 4.89	67.08 ± 5.05	70.36 ± 4.78	0.798
First-month weight	68.88 ± 3.40	74.28 ± 4.64	82.32 ± 5.03	0.155
Second-month weight	72.40 ± 4.05 ^b^	83.16 ± 4.35 ^ab^	88.44 ± 3.83 ^a^	0.046
Third-month weight	75.34 ± 4.47 ^b^	88.66 ± 4.67 ^a^	94.10 ± 3.01 ^a^	0.02
Final weight	73.72 ± 4.76 ^b^	91.88 ± 5.19 ^a^	96.76 ± 3.17 ^a^	0.008
ADG	0.07 ± 0.05 ^b^	0.21 ± 0.05 ^a^	0.22 ± 0.03 ^a^	0.027
TWG	7.92 ± 5.60 ^b^	24.80 ± 6.04 ^a^	26.40 ± 3.57 ^a^	0.027

^1^ Data are expressed as mean ± SEM. LP, low protein level; MP, medium protein level; HP, high protein level. ^2^ ADG, average daily gain; TWG, total weight gain. Note: In the same group of data, different lowercase letters showed significant differences (*p* < 0.05).

## Data Availability

The mass spectrometry proteomics data have been deposited to the ProteomeXchange Consortium (http://proteomecentral.proteomexchange.org, accessed on 11 August 2023) via the iProX partner repository with the dataset identifier PXD044508.

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
