# Peer review of "Proteomic Analysis Reveals the Effects of Different Dietary Protein Levels on Growth and Development of Jersey-Yak"

_animals, 2024, doi:10.3390/ani14030406_

Round 1
Reviewer 1 Report
Comments and Suggestions for Authors
The introduction effectively outlines the significance of the study and provides a clear rationale for investigating the effects of different dietary protein levels on Jersey-yak. Consider briefly expanding on the current knowledge gap and potential implications for the livestock industry.Regarding the methods employed, the author diligently applied TMT quantitative techniques to discern variations in protein expression associated with diverse protein-content diets. A thorough systematic analysis and comparison were carried out. Nonetheless, the description of TMT quantification methods and data analysis lacks the necessary level of detail, thereby compromising the data's credibility. Given that TMT quantification is a pivotal aspect of the article, it is imperative that the author addresses the following inquiries to enhance the clarity and reliability of the methodology section.
1.Page 3, line 103: The methods section mentions the use of 18 samples divided into three groups, implying six biological replicates per group. However, both PCA and subsequent analyses suggest only three biological replicates in each group. The author should provide clarification on sample utilization. If 18 samples were used and a 10-plex TMT was employed, it is crucial to explain how batches of TMT were calibrated across different sets.
-
2.The author's distinction between "identification" and "quantification" is not sufficiently clear. The author should carefully review and specify whether each protein can be identified or quantified. Given the inclusion of biological replicates in each group, it is essential to clarify the handling of missing values, details of data normalization, and any calibration processes undertaken.
-
3.The BCA assay may be susceptible to interference from certain chemicals, particularly reducing agents like DTT. Considering the potential absence of DTT in the lysate, the accuracy of the author's BCA results may be compromised. This should be acknowledged and addressed in the discussion of the results.
Some minor suggestions:
-
1.Page 3, lines 140-141: There is a discrepancy in the starting dose mentioned, with uncertainty between 200ug and 300ug per group. The author should provide clarification and unify the reported starting dose for transparency and accuracy.
-
2.How did the author conduct the peptide assay before TMT labeling? Elaborate on the specific steps and procedures involved in the peptide assay to enhance the reproducibility of the study.
-
3.It is recommended that the author explicitly state the type of TMT kit used, whether 10-plex or 16-plex, on Page 4, line 157. This information is crucial for readers to understand the experimental setup.
-
4.Page 4, line 182: The use of an isolation window of 1.2 may lead to coalescence. The author should address this potential issue and provide detailed information on the data processing steps undertaken to mitigate such challenges, thereby increasing the credibility of the data.
-
5.Page 4, line 185: The phrase "he final LC-MS" may need correction to "The final LC-MS" for grammatical accuracy.
Reviewer 2 Report
Comments and Suggestions for Authors
This study evaluated supplementation of crude protein on growth of Jersey-yak hybrids. Authors used many different analyses of the proteomic data to indicate that there were differences in protein expression related to growth, function, and cellular processes.
The analysis of the proteome data was complete and perhaps went beyond what was needed to show the impacts of the treatments on differentially expressed proteins. It is not clear as the reason for the need to conduct proteomic analysis to answer the objective of the study, which was to improve production of Jersey-yaks through supplementing a grazing diet.
As the LP treatment consisted only of grazing intake, information about the grazing land is important, particularly the time of the growing season and the state of the grass that was grazed, as protein levels would vary greatly throughout the year. As the LP group lost weight in the last month, was it due to the time of year and amount of forage that was available for grazing?
Also, the supplementary diet would have caused a net increase in energy consumption as well as protein. How is it known if the changes are due to increased protein or to increased total energy? There is not any information about how the supplementary diets were provided to the Jersey-yaks. Were they individually fed or group fed? Did the animals eat the entire supplemental diet each day?
The only growth data provided was weight and average daily gain. For this type of study, indications of changed carcass composition with greater muscle would make the study much stronger. Were any carcass measurements taken such as LD muscle area cross-section?
In the materials and methods, there is a lot of abbreviation use that is not defined. DTT, IAA, TFA, TEAB, FA, PMSF
For western blotting, what was the secondary antibody? What was the detection solution (chemiluminescence)? What imaging was used to acquire images?
What was the buffer concentrations for the NH4HCO3 buffer?
Please convert rpm to x g for centrifugation speeds as rpm is not a consistent measurement of force or speed between centrifuge models.
Line 185 – missing a T.
Figure 2. Need more description of the figures in the legend. What is shown in the volcano plots? What do the numbers in the Venn diagram indicate? The cluster analysis - is it showing individual animals?
Discussion of the cluster analysis should be expanded and clarified. What is an abundance value? Is it a fold-change from a baseline?
L289-291 – this sentence needs clarity, what about this cluster analysis shows growth and development? Unless the DEPs are identified as related to growth and development, this statement is over
What is the difference in cluster analysis between Figure 2C and Figure 5? Are both needed to show the results of the study?
The cluster analysis figures are blurry and hard to read the small print.
The differences in comparison between LP and HP vs LP and MP is interesting. The DEPs and groups of DEPs that are identified seem to show more active growth happening in MP whereas the HP group seemed to have decreased activity of remodeling that is seen with muscle growth, perhaps indicated a shift away from muscle growth towards other tissue growth? This also seems apparent in the K-means cluster map. It also appears that the LP group, because they were losing weight, may be going through muscle catabolism. The discussion includes a break down of individual proteins and systems that are differentially expressed and relates them to muscle development.
Lines 512-520 – the discussion about hypoxia-related pathways should be made clearer. What was the difference between the treatment groups and why would it be important?
Comments on the Quality of English LanguageEnglish quality is fine. There are a few statements that need clarity, but the manuscript was easily read.
Reviewer 3 Report
Comments and Suggestions for Authors
The manuscript "Proteomic analysis reveals the effects of different dietary protein levels on growth and development of Jersey-yak" is a well-written manuscript. The authors were interested in identifying novel, readily easy methods of determining the impact of changes in the feeding system on growth traits of Jersey-yak under different protein levels. While the sample numbers are on the lower side, this data does provide the research community with novel traits to peruse and warrants publication in Animals.
Specific points:
L62 and throughout the manuscript: Use as the Longissimus thoracis et lumborum (LTL) or to either of its two parts, Longissimus thoracis (LT) or longissimus lumborum (LL), depending on which is referenced. The word dorsi should be omitted.
L120: The feeds and chemical composition of diets (LP, MP and HP) should be reported as regular Table (e.g., Table 1).
L141-154: This description appears to be an lab protocol. Please, rewrite.
L185: "The"
L208 and Figure 7: Western blotting assay and its results should be reported as supplementary materials.
L242: There is no need to report monthly weigh gain. Please, remove.
L279: Correct the caption of your Figures.
L295: Consider to report the BPs, MFs and CCs by using Metascape.
L572-573: Remove economic benefits and "According to the results of proteomics analysis, a total of 434 DEPs were identified."
Comments on the Quality of English LanguageMinor editing of English language required.
Round 2
Reviewer 1 Report
Comments and Suggestions for Authors
Agree to accept this version
Author Response
We thank the reviewer for this insightful comment. We thank the reviewers for recognising our work.
Reviewer 3 Report
Comments and Suggestions for Authors
Dear authors, before the final acceptance, please have a look in the following paper: https://doi.org/10.1016/0309-1740(90)90010-4
Briefly, considering the portion of the longissimus used by most meat and muscle scientists includes only the thoracis or lumborum or both, use only these designations and abbreviations for identification. Once the designation has been introduced, the abbreviation should be used thereafter. Here are our specific recommendations: longissimus thoracis et lumborum (LTL) longissimus thoracis (LT) longissimus lumborum (LL). The word "dorsi" should be omitted.
Thanks.
